# A Comparative Pharmacokinetic Study for Cysteamine-Containing Eye Drops as an Orphan Topical Therapy in Cystinosis

**DOI:** 10.3390/ijms25031623

**Published:** 2024-01-28

**Authors:** Anita Csorba, Gábor Katona, Mária Budai-Szűcs, Diána Balogh-Weiser, Péter Molnár, Erika Maka, Adrienn Kazsoki, Márton Vajna, Romána Zelkó, Zoltán Zsolt Nagy, György T. Balogh

**Affiliations:** 1Department of Ophthalmology, Semmelweis University, Mária Street 39, H-1085 Budapest, Hungary; 2Institute of Pharmaceutical Technology and Regulatory Affairs, Faculty of Pharmacy, University of Szeged, Eötvös Street 6, H-6720 Szeged, Hungary; 3Department of Physical Chemistry and Materials Science, Faculty of Chemical Technology and Biotechnology, Budapest University of Technology and Economics, Műegyetem rkp. 3, H-1111 Budapest, Hungary; 4Molteam Llc., Mélyfúró Street 4, H-1151 Budapest, Hungary; 5University Pharmacy Department of Pharmacy Administration, Semmelweis University, Hőgyes Endre Street 7-9, H-1092 Budapest, Hungary; 6Department of Pharmaceutical Chemistry, Semmelweis University, Hőgyes Endre Street 7-9, H-1092 Budapest, Hungary

**Keywords:** cystinosis, cysteamine, corneal permeability, physicochemical characterization, in vitro and ex vivo study

## Abstract

Cystinosis is a low-prevalence lysosomal storage disease. The pathomechanism involves abnormal functioning of the cystinosine lysosomal cystine transporter (CTNS), causing intraliposomal accumulation of the amino acid cysteine disulfide, which crystallizes and deposits in several parts of the body. The most common ophthalmic complication of cystinosis is the deposition of “gold dust” cystine crystals on the cornea, which already occurs in infancy and leads to severe photosensitivity and dry eyes as it gradually progresses with age. In the specific treatment of cystinosis, preparations containing cysteamine (CYA) are used. The availability of commercialized eyedrops for the targeted treatment is scarce, and only Cystadrops^®^ are commercially available with strong limitations. Thus, magistral CYA-containing compounded eyedrops (CYA-CED) could have a key role in patient care; however, a rationally designed comprehensive study on the commercialized and magistral products is still missing. This work aims to build up a comprehensive study about commercialized and magistral CYA eye drops, involving pharmacokinetic and physicochemical characterization (applying mucoadhesivity, rheology test, investigation of drug release, and parallel artificial membrane permeability assays), as well as ex vivo tests, well supported by statistical analysis.

## 1. Introduction

Cystinosis is a rare, autosomal recessively inherited lysosomal storage disease. It is characterized by a mutation of the cystinosin lysosomal cystine transporter (CTNS) gene located at chromosome 17p13, which encodes cystinosin lysosomal membrane transporter protein. In the case of abnormally functioning CTNS, the disulfide of the amino acid cysteine (cystine) accumulates intralysosomally, crystallizes, and becomes deposited throughout the body, causing progressive functional impairment of the organs [1]. The most common form of the disease is the infantile nephropathic form, which presents in about 95% of cases and is characterized by the most severe course [2]. The initial symptoms might develop in infancy, commonly in the form of renal failure. In addition to the kidneys, the ocular structures are affected the most [3]. Ophthalmologic symptoms are mostly caused by the corneal deposition of “gold-dust” cystine crystals. Crystal deposition occurs during infancy and progresses gradually with age, leading to severe photosensitivity, blepharospasm, and dry eyes [4].

In the specific treatment of cystinosis, cysteamine-containing preparations are used. Cysteamine can dissolve accumulated intracellular cystine crystals by a disulfide interchange reaction, forming cysteine-cysteamine disulfide [5]. Due to its similar structure to amino acid lysine, excretion of this metabolic compound is obtained through the excretion pathway of lysine and does not require the cystinosine transporter. Systemic accumulation of cystine crystals can be effectively treated with orally administered cysteamine, however, it does not reduce corneal crystal deposition due to its low biodistribution into the avascular corneal tissue [6]. The ophthalmic treatment includes the administration of topical eye drops containing cysteamine hydrochloride (CYA). The conventional formulation of the eye drops is an aqueous solution of CYA [7]. According to the standard clinical recommendations, it requires continuous and frequent instillation from six to twelve times daily [8]. At room temperature, the active substance cysteamine oxidizes to a clinically ineffective form, thus, continuous refrigeration is needed to maintain stability. All of these factors may add up to a weakening of therapeutic compliance on the part of the patients. Another disadvantage is the fact that Cystadrops^®^ are only available as an orphan drug since 2008 in the European Union; to overcome this problem, an alternative ocular formulation of magistral eye drops are prepared with various concentrations of CYA, using different excipients, and buffers [4,9].

Recently, the efficacy of aqueous CYA topical eye drops in reducing corneal cystine crystals has been controversial [1,10]. The corneal epithelium represents a mainly lipophilic diffusion barrier against hydrophilic agents, such as CYA. In addition, a significant portion of the aqueous solution leaves the eye surface through the lachrymal drainage. Thus, various drug delivery systems have been developed in order to increase stability, ocular retention, and prolonged release of cysteamine, such as hydrogels, contact lenses, and nanowafer discs [11,12,13]. Recently, a newly formulated viscous, CYA-containing topical preparation (Cystadrops^®^, 0.55% cysteamine hydrochloride, Orphan Europe, Puteaux, France) has become available. This preparation is stable at room temperature and the dose regimen is less frequent with four times of daily instillations, which results in better patient compliance [9,14]. Recent data showed that Cystadrops^®^ are superior to the aqueous CYA formulation in terms of reducing the amount of corneal crystals in vivo [9,14]. It is hypothesized that the newly introduced bioadhesive solution prolongs the corneal contact time due to its viscous consistence; hence, it may improve drug absorption and bioavailability. However, this has not been proven yet since the distribution of cysteamine in ocular tissues during the application of the aqueous CYA formulation or Cystadrops^®^ has not been investigated by in vitro and ex vivo experiments.

This study is aimed to fill a big gap about the properties of commercial eye drop Cystadrops^®^ and a CYA-containing compounded eye drop (CYA-CED). A detailed comprehensive study had been elaborated to characterize and compare the pharmacokinetic properties of CYA-CED and Cystadrops^®^ using physicochemical, in vitro, and ex vivo test systems. First, the osmolality, viscosity, and dissolution profile of two formulations were evaluated. Corneal permeability and retention were also analyzed using an in vitro corneal parallel artificial membrane permeability assay (corneal-PAMPA) and an ex vivo porcine cornea. The well-designed data set about the cysteamine-containing eyedrops could introduce a novel methodical approach in the pre-clinical steps of ophthalmic drug development (Figure 1).

## 2. Results

### 2.1. Physicochemical Parameters of Eye Drops

As representative physicochemical parameters of eye drops, pH values, osmolarity and viscosity values were determined to compare the different CYA-containing eye drops.

The pH of the CYA-containing compounded eye drop (CYA-CED) and Cystadrops^®^ eye drop references were 7.60 and 5.22, respectively. The CYA-CED showed a value close to physiological tear osmolarity of 282 ± 1 mOsm/L, while the Cystadrops^®^ had higher osmolality (338 ± 1 mOsm/L). The compounded eye drops also meet the physiologically tolerated and ideal range requirements. The pH range that is still physiologically tolerated is 4.5–8.5 (ideally 6.5–7.9), and 220–430 mOsmol/kg (ideally 270–320 mOsmol/kg) for osmolality.

The viscosity of Cystadrops^®^ was orders of magnitude greater than that of CYA-CED, which can be explained by the different types and concentrations (5.2% *w*/*w* carboxymethylcellulose and 0.39% *w*/*w* hydroxyethylcellulose) of polymers (Figure 2). However, our main expectation was the implementation of sterile filtration regarding CYA-CED as a compounded eye drop, which is only possible in the case of low viscosity. Moreover, highly viscous eye drops (>55 mPa) may increase reflex tearing, thus causing a faster drug removal from the corneal surface; they were found to be irritating for several patients, hindering reproducible drug dosing; and, in addition, they can cause blurred vision following administration. Furthermore, in the case of formulations with lower viscosity, more uniform droplets can be formed during administration [15,16]. The compounded eye drop viscosity corresponds to the tear film. Moreover, the high viscosity of formulation can hinder drug release, and therefore decrease the permeability rate of the drug unless the polymer used has mucoadhesive properties, which is a pivotal point.

### 2.2. Comparison of the Mucoadhesivity

The eye is more exposed to various elimination mechanisms, so after using an eye drop, one has to expect rapid precorneal elimination, which significantly reduces the bioavailability of the preparation; thus, traditional eye drops have approximately 2–10% bioavailability. The residence time of the eye drops can be increased with the increase of the viscosity of the eye drops or mucoadhesive formulations.

In our case, the mucoadhesive behavior of the CYA-CEDs and the reference formulation Cystadrops^®^ were compared. Based on our results, the referenced eye drops had significantly higher mucoadhesivity in vitro. A significantly higher adhesive force and adhesive work was measured in the case of the Cystadrops^®^ (Figure 3A,B). Interestingly, there is a more than 10-fold higher concentration difference between the two polymers (HEC and CME: CYA-CED and Cystadrops^®^, respectively), however, the mucoadhesive force is only about two times higher, whereas the mucoadhesive is work about three times higher, which may call into question the necessity of the relatively high polymer content of Cystadrops^®^. Lower carboxymethylcellulose content may have resulted in similar mucoadhesive superiority, but lower viscosity, and therefore more uniform dosage. However, based on the results, longer residence time of the Cystadrops^®^ can be expected in vivo.

### 2.3. In Vitro Drug Release Study

The RED method was applied to compare the drug release of Cystadrops^®^ to CYA-CED. The time-dependent drug release profiles can be seen in Figure 4.

The CYA-CED shows a burst-like drug release, whereas Cystadrops^®^ indicate rather diffusion-controlled CYA release, which can be explained by the higher viscosity and mucoadhesivity of Cystadrops^®^. In the case of the compounded eye drop, a significantly higher drug release rate was observed in the first 60 min in comparison to Cystadrops^®^, which assumes faster corneal absorption. Drug release on the ocular surface is a crucial issue in reaching the desired steady-state concentrations in the aqueous humor, because only the released or dissolved drug can be absorbed into the eye. Furthermore, within 4 h more than 90% of drug content was released from both formulations, which, together with the appropriate mucoadhesive property, also supports long-lasting therapeutic effects.

### 2.4. In Vitro Corneal Permeability Study

PAMPA measurement results showed an opposite tendency in corneal permeability of the CYA-containing formulations (Figure 5A). In the case of Cystadrops^®^, significantly higher effective permeability values were obtained. The identified difference in permeability and membrane retention between the two formulations can be traced back to two differences resulting from the physicochemical properties that were verified during the characterization of the eye drop forms. Firstly, the pH value of the two forms differs, which, taking into account the amphoteric nature of CYS and its related proton dissociation constants (pKa(-SH) = 8.34, pKa(-NH_2_) = 10.88), may affect the permeability of CYS [17]. In this case, in the eye drop medium provided by the lower pH value of Cystadrops^®^, the thiol is 100% neutral, while the primary amine group is completely ionic. On the contrary, in the case of CYS-CED, the ionic nature of the NH_2_ group does not change in the more basic medium (pH = 7.6), but the SH group can become partially ionic (based on the Henderson–Hasselbalch equation—S^−^: 15.4%), which can reduce the lipophilic character of CYS and, in parallel, its permeability and membrane retention. In addition, the higher osmolality and electrolyte concentration of Cystadrops^®^ can also be considered as factors affecting permeability. According to the ion-pairing partition effect described by Alex Avdeff, the lipophilicity of the ionizable, in this case zwitterionic, CYS compound can increase due to the increased electrolyte concentration, which can also increase the permeability and membrane retention capacity of CYS [18].

Similarly, when calculating the flux (Equation (3)) and membrane retention (Equation (2)), Cystadrops^®^ showed significantly higher flux and membrane retention in comparison to the compounded eye drop (Figure 5B).

### 2.5. Ex Vivo Corneal Permeability on Porcine Eyes

In the ex vivo corneal permeability study, concentrations of CYA were measured at 15, 30, and 60 min after instillation in the precorneal area, cornea, and aqueous humor (Figure 6).

As the results show, a higher CYA concentration remained in the precorneal area in the case of the compounded eye drop (CYA-CED) under the 60 min treatment (although not significant), which suggests that Cystadrops^®^ penetrated with higher tendency into the cornea (Figure 6A). CYA concentrations measured in the porcine cornea support our former observation; at all time points, CYA permeation from Cystadrops^®^ was higher in comparison to the compounded formulation, moreover, at 15 min this difference was even more significant (Figure 6B). This may be related to the EDTA content of Cystadrops^®^, as a well-known penetration enhancer. As a chelating agent, EDTA is able to disrupt the tight junctions and adherent junctions by sequestration of interstitial Ca^2+^ ions, on which the barrier function is dependent, and distribute them in the cornea [19]. Interestingly, in the aqueous humor, the penetration of CYA from compounded preparations was higher at all time points, especially at 15 min, where a significant difference was achieved (Figure 6C). This can be explained by the lower viscosity and weaker mucoadhesion, resulting in weaker secondary CYA-polymer interaction, which increases the transport of CYA into the aqueous humor without persistent corneal distribution.

Based on the determined CYA concentration values, permeability toward the corneal tissue and the aqueous humor, as well as corneal retention, was calculated using Equations (4)–(6) and plotted in Figure 7. In accordance with the determined CYA concentrations, toward the corneal tissue, Cystadrops^®^ showed significantly higher permeability than the CYA-CED at 15 min; after that, this difference decreased, but the superiority of Cystadrops^®^ remained throughout the 60 min treatment (Figure 7A). CYA permeability of CYA-CED was higher toward the aqueous humor in the first 30 min (only significant at 15 min), however, this difference between the two formulations decreased and eventually reached equilibrium after 60 min (Figure 7B). In the case of corneal retention, an insignificant difference was observed between the two formulations with the superiority of Cystadrops^®^, which can be explained by the higher viscosity, mucoadhesion, and the effect of carboxymethylcellulose, which binds to corneal epithelial cells and remains bound for at least several hours, resulting in sustained corneal permeation (Figure 7C) [20].

## 3. Discussion

The marketing authorization of Cystadrops^®^ was validated in 2017 by the European Medicines Agency for treatment of corneal manifestation of cystinosis among children older than 2 years of age [7]. Since then, the safety and the effectiveness of Cystadrops^®^ in dissolving corneal cystine crystals and reducing photosensitivity in both adults and children has been demonstrated [9,14]. Following the application of Cystadrops^®^ four times daily for a period of 30 days, a significant reduction in the density of crystal deposition and improvement in photosensitivity was observed. Subsequently, a plateau was observed in both the quantity of crystals and the severity of photosensitivity after an initial gradual decrease within the first year, allowing for a reduction in the frequency of application [13]. The 0.55% viscous solution formulation of Cystadrops^®^ was found to significantly reduce the extent of crystal deposition compared to the 0.1% aqueous cysteamine drops [9]. Additionally, a randomized multicenter study has also indicated a decrease in crystal density, highlighting the improved efficacy of the viscous formulation compared to the 0.1% cysteamine-containing solution [21]. Some case reports have also reported significant therapeutic effects of Cystadrops^®^ in terms of decreasing photosensitivity and crystal density [5,22]. According to the clinical findings, it is plausible to hypothesize that the viscous formulation allows for prolonged retention of the active substance on the ocular surface, leading to elevated concentrations within the corneal layers, which, in turn, results in an enhanced dissolution of cystine crystals. Both our in vitro and ex vivo investigations substantiated our hypotheses. Our measurements showed that the elevated viscosity and inherent mucoadhesive property of Cystadrops^®^ facilitates an extended residence time on the precorneal surface and enables a diffusion-controlled release of CYA. Moreover, both in vitro corneal permeability and ex vivo corneal CYA concentration and retention were higher in the case of Cystadrops^®^ compared to the CYA-CED solution. Consequently, due to these physicochemical and pharmacokinetic attributes, Cystadrops^®^ could be considered superior to the aqueous cysteamine solution for the ocular management of cystinosis. On the other hand, considering that a significant difference could be identified between the ex vivo results of the two eye drop formulas only in the first kinetic point, the use of compounded eye drops may be an alternative to be considered in case of difficulty in supplying Cystadrops^®^. Compounded formulations with several components can be prepared in the pharmacy following the official pharmacopoeial regulations in accordance with the principles set out in the resolution issued by the Council of Europe on the quality and safety assurance requirement for medicinal products prepared in pharmacies for the special needs of patients. If the active ingredients are available in the official pharmacopeia or the Authority provided permission for its particular use, compounding can be a promising alternative to substitute the medicine shortage.

Enhancing the expanded production of compounded formulations beyond the scope of Good Manufacturing Practice (GMP) regulations is possible by establishing a risk-based framework. This framework should involve the direct oversight of healthcare professionals from Health Authorities, specifically focused on the circumstances of the manufacturing steps, including analytical support and manufacturing validation, and the physicians who should also approve therapeutic protocols before approving these medicines. This risk-based framework should be well-defined and regulate potential patient population (individual therapy planning) and production circumstances.

It must be admitted that producing medicinal products outside of GMP puts a higher risk on the final products; however, the absolute lack of a particular product can put a higher risk on patients instead.

## 4. Materials and Methods

### 4.1. Chemicals

Cysteamine (CYA) was purchased from TCI (>98.0%, Tokyo, Japan). L-α-phosphatidylcholine, mucin from a porcine stomach (Type II), sodium dodecyl sulphate, phosphate buffer (PBS, pH 7.4), as well as components for simulated tear fluid (STF) were purchased from Sigma-Aldrich/Merck (Budapest, Hungary). The analytical grade solvents including methanol, acetonitrile, hexane, dodecane, and chloroform were purchased from Merck KGaA (Darmstadt, Germany). A sample of 0.55% viscous CYA eye drop (Cystadrops^®^, 3.8 mg/mL cysteamine hydrochloride) was obtained from Recordati Rare Diseases, (Puteaux, France).

### 4.2. Preparation of Compounded Eye Drops

In comparison to the referenced Cystadrops^®^, a compounded eye drop formulation containing the same amount of CYA was prepared from the available compounds that are officially on the positive list of the National Institute of Pharmacy and Nutrition, Hungary (Table 1). Along with being on the positive list, hydroxyethyl cellulose (viscosity: 250–400 mPa·s, Ph.Hg. VIII) acts as a demulcent by relieving eye inflammation, irritation, and dryness [23]. The applied polymer concentration enabled sterile filtration. The preparation of the solution was conducted in a laminar air flow cabinet under aseptic conditions consisting of four main steps: (1) preparation of the buffer solution, (2) mixing of hydroxyethylcellulose into a portion of the buffer solution to produce a viscous solution, (3) dissolution of the active substance and other excipients in the remaining buffer solution, and (4) mixing of the two solutions followed by sterile filtration through a sterile polyethersulfone (PES) filter membrane of 0.22 µm pore diameter and 25 mm diameter (Stericup^®^, Merck-Millipore, Darmstadt, Germany).

### 4.3. Characterization of Eye Drops

The pH of ophthalmic formulations plays an important role in a patient’s comfort during application. The pH of the eye drops was measured using a WTW^®^ inoLab^®^ pH 7110 laboratory pH-tester (Thermo Fisher Scientific, Budapest, Hungary). A freezing point depression-based osmometer was used for osmolarity detection (KNAUER K-4700S Semi-Micro Osmometer, Berlin, Germany). The samples were stored in a refrigerator at 2–8 °C until the measurement was performed. All measurements were performed in triplicate (n = 3). Data were expressed as means ± SD. The viscosity of the eye drops was measured using an Anton Paar Physica MCR301 rheometer, and temperature was kept at 35.0 ± 0.1 °C using a Peltier device during all measurements. A cone-plate probe with a diameter of 25 mm (CP25-1) was applied, and 100 µL formulations were placed on the lower plate. The measurement was carried out in a frequency range of 1–1000 1/s in triplicate.

### 4.4. Investigation of the Mucoadhesivity

The mucoadhesive properties of the formulations were investigated by using a TA.XT plus Texture Analyzer (Stable Micro Systems Ltd., Surrey, UK). A 5 kg load cell was used. A filter paper wetted with 50 µL of 8% (*w*/*w*) mucin dispersion was fixed into the mucoadhesion test rig. The mucin dispersion was prepared with pH 7.4 STF (6.78 g/L NaCl, 2.18 g/L NaHCO3, 0.084 g/L CaCl_2_ × 2H_2_O, 1.38 g/L KCl in purified water). Twenty of each sample were placed onto the cylinder probe with a diameter of 10 mm. A 2500 mN preload was used for 3 min, then the probe was moved upwards at 2.5 mm/min speed. The force–distance curve was recorded. In order to characterize the mucoadhesive behavior, the adhesive force (mN) and work of adhesion (mN mm) were used; the latter was calculated from the area under the force–distance curve. Five parallel measurements were carried out.

### 4.5. Rapid Equilibrium Dialysis (RED) Experiments

The time-dependent release profile of Cystadrops^®^ and CYA-containing compounded eye drops (CYA-CED) was investigated using the RED Device (Thermo Scientific™, Waltham, MA, USA). The RED Device inserts (8K MWCO) were fitted into a reusable Teflon base plate, and then 150 µL of formulations were placed into the donor chambers. Thereafter, 300 µL of phosphate buffer (pH 7.4) was added to the acceptor chambers, and the unit was covered with a sealing tape and incubated at 35.0 °C on an orbital shaker at 350 rpm for 4 h. Samples were withdrawn at 15, 30, 60, 120, and 240 min from the acceptor chambers and CYA concentrations were determined using HPLC-DAD. Five parallel measurements were performed, and the data are presented as mean ± SD.

### 4.6. High Performance Liquid Chromatography (HPLC)

Quantification of CYA concentration in the experiments was performed using an Agilent 1260 (Agilent Technologies, Santa Clara, CA, USA) HPLC. At the stationary phase, a Kromasil^®^ C18 column (5 µm, 250 mm × 4.6 mm (Phenomenex, Torrance, CA, USA) was applied. The mobile phase consisted of 620 mL water, 330 mL acetonitrile, 50 mL methanol, 1.4 mL orto-phosphoric acid (85% *v*/*v*), and 11.52 g of sodium dodecyl sulphate. The isocratic elution was performed for 10 min at a flow rate of 1.0 mL/min at 50 °C [25]. Chromatograms were detected at 210 nm using a UV–VIS diode array detector. Data were evaluated using ChemStation B.04.03. Software (Agilent Technologies, Santa Clara, CA, USA). Retention time was 4.12 min, and linearity was between 10 µg/mL and 1000 µg/mL. The limit of detection (LOD) was 234 ng/mL, whereas the limit of quantification (LOQ) was 709 ng/mL.

### 4.7. In Vitro Corneal Permeability Measurements

In vitro transcorneal permeability of CYA-containing formulations was investigated using a corneal-specific parallel artificial membrane permeability assay (corneal-PAMPA) developed by us [26]. The filter donor plate (Multiscreen™-IP, MAIPN4510, pore size 0.45 µm; Millipore, Merck Ltd., Budapest, Hungary) was coated with 5 µL of lipid solution (16 mg phosphatidylcholine dissolved in a 600 µL solvent mixture (70% (*v*/*v*) hexane, 25% (*v*/*v*) dodecane, 5% (*v*/*v*) chloroform). The Acceptor Plate (MSSACCEPTOR; Millipore, Merck Ltd., Budapest, Hungary) was filled with 300 μL of PBS (pH 7.4). Then, 150–150 μL of the formulations were transferred to the membrane of the donor plate. Then, the latter was covered with a plate lid in order to decrease the possible evaporation of the solvent. This sandwich system was incubated at 35.0 °C for 4 h. The concentration of CYA permeated in the acceptor plate was determined by HPLC. The effective permeability and membrane retention of CYA was calculated using the following Equation (2):(1)$$P_{e}=\frac{-2.303}{A\cdot (t-\tau_{ss})}\cdot \left(\frac{1}{1+r_{v}}\right)\cdot \mathrm{lg}\left[-r_{v}+\left(\frac{1+r_{v}}{1-MR}\right)\cdot \frac{c_{D}\left(t\right)}{c_{D}\left(0\right)}\right]$$
where *P*_e_ is the effective permeability coefficient (cm/s), *A* is the filter area (0.3 cm^2^), *t* is the incubation time (s), *τ_ss_* is the time to reach steady-state (s), *r_v_* is the volume ratio of aqueous compartments (VD/VA), VD and VA are the volumes in the donor (0.15 mL) and acceptor phase (0.3 mL), *c_D_*(*t*) is the concentration of the compound in the donor phase at time point t (mol/mL), *c_D_*(0) is the concentration of the CYA in the donor phase at time point zero (mol/mL), and *MR* is the membrane retention factor (%), defined as follows [14]:(2)$$MR=1-\frac{c_{D}\left(t\right)}{c_{D}\left(0\right)}-\frac{V_{A}c_{A}(t)}{V_{D}c_{D}(0)}$$
where *c_A_*(*t*) is the concentration of CYA in the acceptor phase at time point *t* (mol/mL). Based on the effective permeability coefficient and the equilibrium solubility of CYA, the flux (J, mol/cm^2^·s) of formulations can be calculated as follows:(3)$$J=P_{e}\cdot Sol$$
where *Sol* is the concentration of the dissolved CYA at 4 h provided by the RED measurement. Six parallel measurements were performed, and the data are presented as mean ± SD.

### 4.8. Ex Vivo Penetration Test

Fresh porcine eyes were first placed into a PTFE holder which surrounds the whole eyeball except the corneal surface. The holder enables the spreading of a test sample with a volume of 1 mL on the surface of the cornea. The eyeballs were incubated at 35.0 °C. Before using the test sample, the surface of the cornea was incubated with a physiological saline solution and then removed immediately before sample insertion. The examined samples were diluted 10 times before the measurement, after which 1 mL of their volume was pipetted to the surface of the cornea. After 15, 30, and 60 min, diluted formulations were removed from the corneal surface, the eye holder cells were disassembled, then the aqueous humor was aspirated through corneal paracentesis using a 25 gauge needle; finally, the cornea was isolated and extracted with methanol:water with the ratio of 1:1 using an orbital shaker (PSU-10i Orbital Shaker, Grant Instruments Ltd., Cambs, England) for 60 min, at 450 rpm. The CYA content of the precorneal fluid, aqueous humor, and the corneal extract was analyzed by HPLC. The corneal retention (CR) of CYA, the apparent permeability (PappC) of CYA into the cornea, and the apparent permeability (PappAq) of CYA into the aqueous humor were calculated by using Equations (4), (5), and (6), respectively:(4)$$CR=1-\frac{c_{CS}\left(t\right)}{c_{CS}\left(0\right)}-\frac{V_{AC}c_{Aq}(t)}{V_{CS}c_{CS}(0)}$$
(5)$$P_{app}C(cm/s)=\frac{∆{\left[C\right]}_{C}\times V_{AC}}{A\times {\left[C\right]}_{CS}\times ∆t}$$
(6)$$P_{app}Aq(cm/s)=\frac{∆{\left[C\right]}_{Aq}\times V_{AC}}{A\times {\left[C\right]}_{CS}\times ∆t}$$
where *c_CS_*(*t*) is the concentration of CYA on the corneal surface at time point t (mol/mL), *c_CS_*(0) is the concentration of CYA on the corneal surface at time point zero (mol/mL), *c_Aq_*(*t*) is the concentration of CYA in the aqueous humor at time point *t* (mol/mL), and VAC and VCS are the volumes in the anterior chamber (0.25 mL) and on the corneal surface (1.05 mL).

PappC was calculated from the concentration difference of CYA in the cornea (Δ[*C*]*C*) after treatment, initial concentration of the compound on the corneal surface at time point zero ([*C*]*CS*), and the volume of anterior chamber VAC (250 µL); A is the surface area available for permeability (1.77 cm^2^), and t is the incubation time (s). PappAq was calculated from the concentration difference of CYA in the aqueous humor (Δ[*C*]*Aq*) after treatment, initial concentration of the compound on the corneal surface at time point zero ([*C*]*CS*), and the volume of anterior chamber VAC (250 µL); A is the surface area available for permeability (1.77 cm^2^), and t is the incubation time (s). Each measurement was performed in triplicate, and the data are presented as mean ± SD.

### 4.9. Statistical Analysis

All data were presented as means ± SD. The statistical significance of the results was investigated with a paired *t*-test or one-way ANOVA with a post hoc test (Tukey’s multiple comparisons test, α = 0.05) using TIBCO Statistica^®^ 13.4 (Statsoft Hungary, Hungary) software. Changes were considered statistically significant at *p* < 0.05.

## 5. Conclusions

A gap-filling study about the cysteamine-containing compounded eyedrops and commercial formulas for the treatment of cystinosis has been performed. The magistral formulas for CYA-CED—following the official pharmacopeial regulations—and Cystadrops^®^, as the only commercially available eye drop, were systematically compared for in vitro physicochemical and ex vivo pharmacokinetic properties. The results show that the physicochemical properties (viscosity, mucoadhesivity, dissolution, and lipid membrane permeability) of Cystadrops^®^ are more favorable than CYA-CED regarding expected pharmacokinetic properties that are relevant in the ophthalmic approach. However, the nearly equivalent ex vivo pharmacokinetic profile obtained in studies on porcine eyes suggests that CYA-CED can serve as a suitable alternative in case of difficulty in supplying Cystadrops^®^, which can reduce the risk of possible interruptions in patient care.

## Figures and Tables

**Figure 1 ijms-25-01623-f001:**
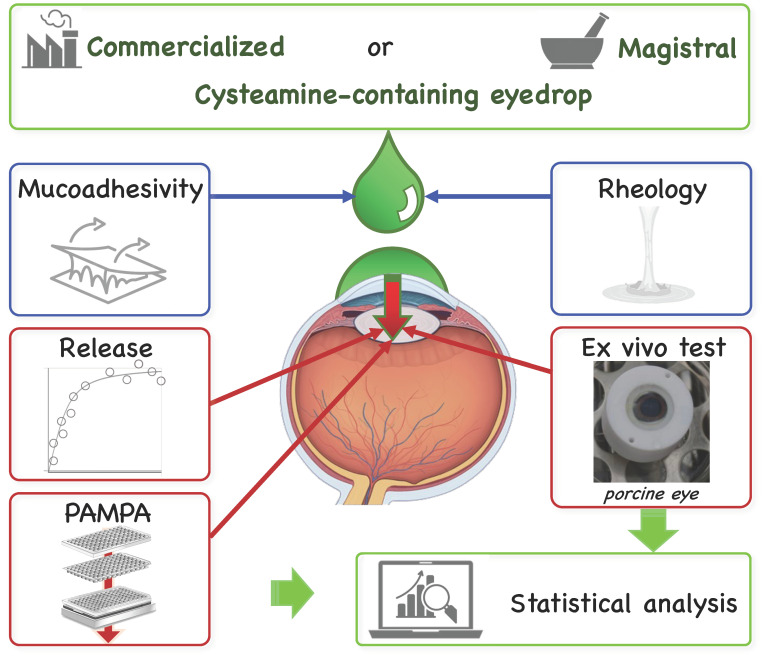
The comprehensive investigation scheme of the cysteamine-containing compounded eye drop (magistral) and Cystadrops^®^ (commercial) formulas (PAMPA—parallel artificial membrane permeation assay).

**Figure 2 ijms-25-01623-f002:**
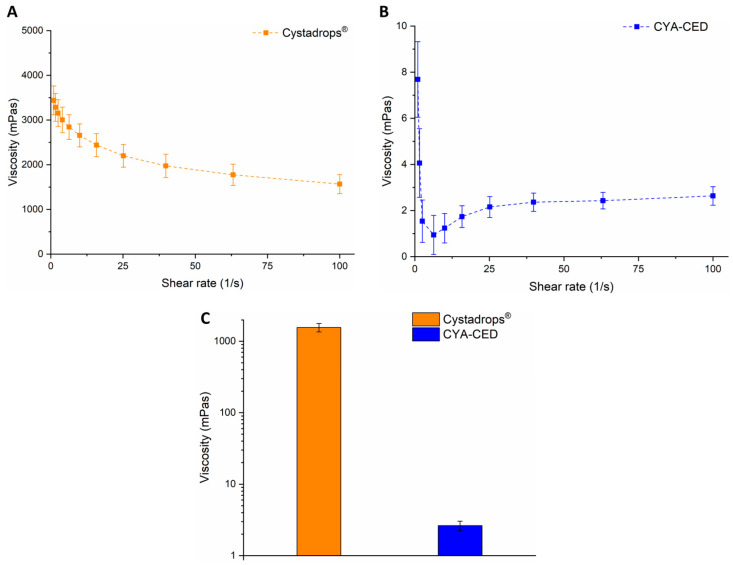
Viscosity of the eye drops: (**A**) Cystadrops^®^ and (**B**) CYA-containing compounded eye drops (CYA-CED) at different shear rates, and (**C**) viscosity of eye drops at shear rate 100 1/s, at 35.0 °C.

**Figure 3 ijms-25-01623-f003:**
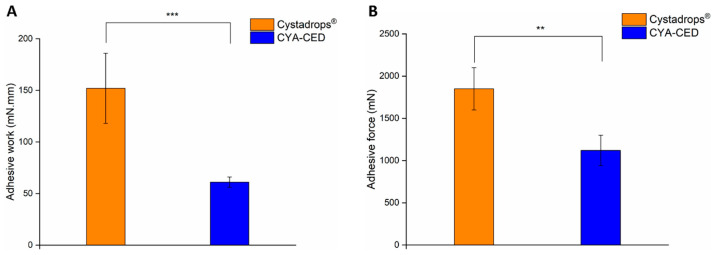
Comparison of adhesive force (**A**) and adhesive work (**B**) of Cystadrops^®^ and the CYA-containing compounded eye drop (CYA-CED). Data are presented as means ± SD, n = 5. (Asterisks indicate significant differences ** *p* < 0.01; *** *p* < 0.001).

**Figure 4 ijms-25-01623-f004:**
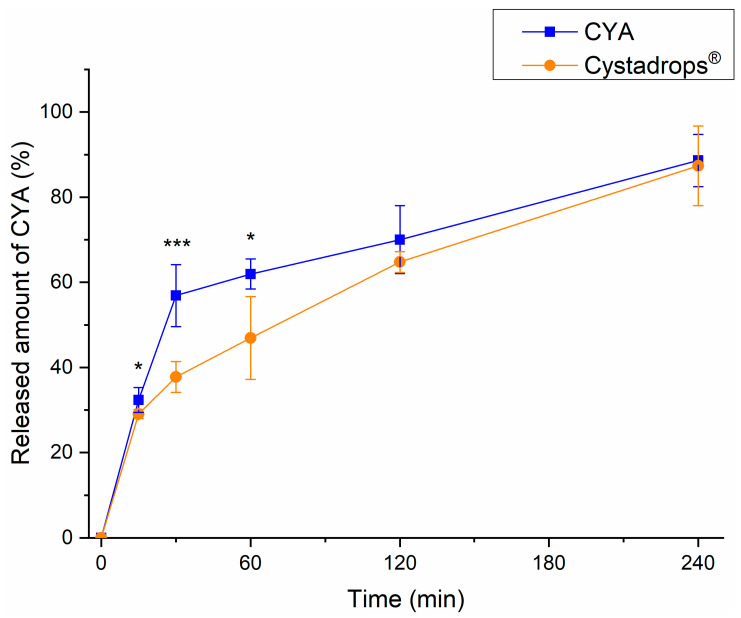
Drug release profiles of Cystadrops^®^ and the CYA-containing compounded eye drop (CYA-CED) determined using the RED method. Statistical analysis: one-way ANOVA with post hoc test (Tukey’s multiple comparisons test, α = 0.05). Data are presented as means ± SD, n = 5. (Asterisks indicate significant differences * *p* < 0.05, *** *p* < 0.001).

**Figure 5 ijms-25-01623-f005:**
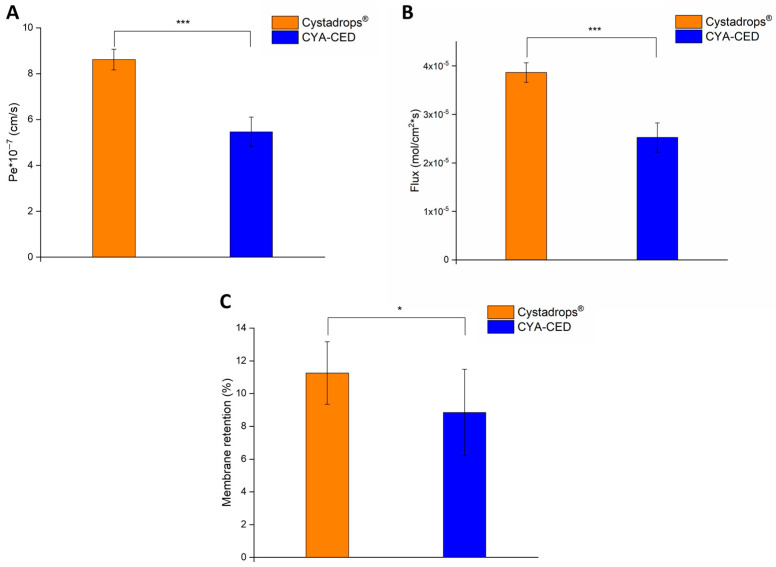
Corneal-PAMPA permeability (**A**), calculated flux (**B**), and membrane retention (**C**) at 4 h of the CYA-containing compounded eye drop (CYA-CED) in comparison to Cystadrops^®^. Statistical analysis: paired *t*-test. Data are presented as means ± SD, n = 6. (Asterisks indicate significant differences * *p* < 0.05; *** *p* < 0.001).

**Figure 6 ijms-25-01623-f006:**
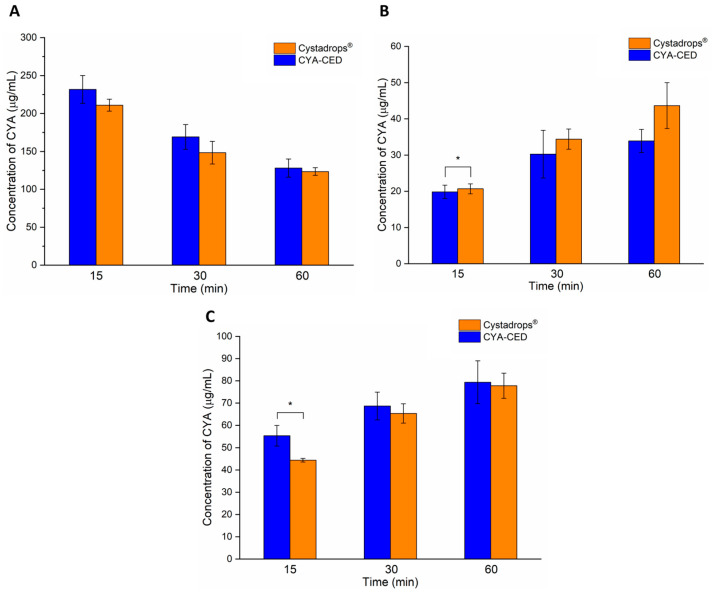
Cysteamine (CYA) concentration measured ex vivo in the different segments of porcine eyes after instillation with the formulations (CYA-containing compounded eye drops (CYA-CED) and Cystadrops^®^) in the precorneal area (**A**), in the cornea (**B**), and in the aqueous humor (**C**). Statistical analysis: paired *t*-test. Data are presented as means ± SD, n = 3. (Asterisks indicate significant differences * *p* < 0.05).

**Figure 7 ijms-25-01623-f007:**
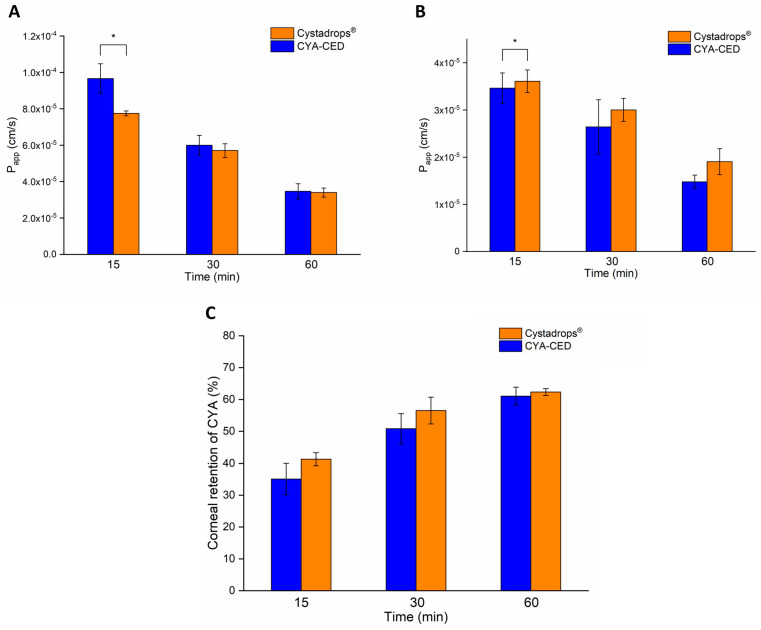
Ex vivo Cysteamine (CYA) permeation towards the cornea (**B**), the aqueous humor (**A**), and corneal retention of CYA (**C**) measured in porcine eyes after instillation with the formulations (CYA-containing compounded eye drop (CYA-CED), Cystadrops^®^). Statistical analysis: paired *t*-test. Data are presented as means ± SD, n = 3. (Asterisks indicate significant differences * *p* < 0.05).

**Table 1 ijms-25-01623-t001:** Composition and dosage form parameters of the CYA-containing compounded eye drop (CYA-CED) versus Cystadrops^®^.

Components/Dosage form Parameters	Function	CYA-CED	Cystadrops^®^ [24]
		Amount (% *w*/*w*)	
cysteamine hydrochloride	API	0.55	0.55
hydroxyethylcellulose	mucoadhesive agent	0.39	
carboxymethylcellulose	mucoadhesive agent	─	5.2
sodium chloride	isotonizing agent	0.29	─
sodium hydrogen carbonate	pH adjustment	0.54	─
citric acid monohydrate	pH adjustment	─	N/A *
disodium edetate	chelator	─	N/A *
hydrochloric acid	pH adjustment	─	N/A *
sodium hydroxide	pH adjustment	─	N/A *
benzalkonium chloride	preservative agent	─	0.01
water for injection	solvent	98.23	N/A *
osmolality (mOsm/L) **		282 ± 1	338 ± 1
pH **		7.60	5.22

* N/A means the marketing authorization holder did not make the data public. ** Data were determined experimentally.

## Data Availability

Data reported in the study are available in the manuscript.

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
