# Peer review of "A Comparative Pharmacokinetic Study for Cysteamine-Containing Eye Drops as an Orphan Topical Therapy in Cystinosis"

_ijms, 2024, doi:10.3390/ijms25031623_

Round 1

Reviewer 1 Report

Comments and Suggestions for Authors

Through an examination of the physicochemical characterization, in vitro and ex vivo performance of Cystadrops® and CYA-containing compounded eyedrops, the authors demonstrated that the higher permeability and mucoadhesivity of Cystadrops® contribute to its better clinical performance. The study design is simple but clear, and the hypothesis is supported by the experimental and calculation results. However, there are a few minor issues that need to be addressed before it can be considered for publication.

1. Were the samples from the two formulations tested under the same CYA dose in the corneal permeability measurement and penetration test? Different doses may influence the results.

2. The drug release study showed that only 240 minutes later, the cumulative drug release of Cystadrops® can reach as high as CYA-CED. Is the Cystadrops® formulation able to retain in the eye for 4 hours? It would be clearer to the audience if the authors could discuss this.

Author Response

Responses to Reviewer #1

Through an examination of the physicochemical characterization, in vitro and ex vivo performance of Cystadrops® and CYA-containing compounded eyedrops, the authors demonstrated that the higher permeability and mucoadhesivity of Cystadrops® contribute to its better clinical performance. The study design is simple but clear, and the hypothesis is supported by the experimental and calculation results. However, there are a few minor issues that need to be addressed before it can be considered for publication.

Response: We thank the Reviewer constructive comments. These are answered point-by-point.

  1. Were the samples from the two formulations tested under the same CYA dose in the corneal permeability measurement and penetration test? Different doses may influence the results.

Response: Both formulations contained the same dose of CYA (0.55% w/w) as shown in Table 1. There was difference only in the excipients.

  1. The drug release study showed that only 240 minutes later, the cumulative drug release of Cystadrops® can reach as high as CYA-CED. Is the Cystadrops® formulation able to retain in the eye for 4 hours? It would be clearer to the audience if the authors could discuss this.

Response: However, both of the formulations contain mucoadhesive excipient, of course, none of the investigated formulations is able to retain in the eye for 240 min, they will be eliminated with blinking ang tearing. The purpose of conducting drug release study for 240 min was the determination of drug release profile and equilibrium solubility of CYA. The most significant difference in drug release was obtained at 30 min, which is relevant from therapeutical point of view, as the residence time of both formulations is expected between 30-60 min therefore, the experienced CYA concentration difference released at 240 min is negligible.

Reviewer 2 Report

Comments and Suggestions for Authors

This is a well written and easy to understand work comparing the main physical and chemical characteristics of two formulations of cysteamine, one commercial and the other prepared as a compounded formulation. Congratulations to the authors for their work in a field where the information available so far is limited, as is often the case with rare diseases.

Most of the abstract is part of the "state of the art", but hardly any information is provided on the content of the paper. Please make reference to the tests that have been performed, as well as the main conclusions of the work.

In line 49, there are two commas instead of a quotation mark. Please correct.

Line 66 states that there are no approved presentations marketed in the European Union, although the Cystadrops presentation is approved in Europe (https://www.ema.europa.eu/en/medicines/human/EPAR/Cystadrops), although it is not marketed in all countries. However, it can be purchased as a foreign drug.

PAMPA meaning in figure 1 should be explained in the caption of line 97.

Please, move the “Material and methods” section before “results”. With this, the material and methods would be item 2, the results would be item 3, and the discussion would be item 4.

One of the two formulations studied is a compounded formulation. Please describe in detail the steps followed for its preparation in the material and methods section. Based on what is the hydroxyethylcellulose percentage of 0.39% chosen? Are special sterile conditions required for its production? Does it start with sterile material? Does it work in laminar flow cabinets?

This compounded formulation, with such a large number of components, is not feasible for routine preparation in pharmacy services. Please discuss this in the discussion section.

Given the high oxidizability of cysteamine, what is the shelf life of the formulation if stored in a refrigerator? In RED and In vitro corneal permeability experiments, temperatures of 35 degrees are used for four hours. If you do not measure cystamine, the main degradation product of cysteamine, how do you know which part of the cysteamine is degraded and, for that reason, its passage through the membrane is not detected?

Lines 107-108 should be part of the discussion, and it should be in this section that we comment on which of the two is more physiological. The text between lines 108-118 clearly corresponds to the discussion section, not the results. The clinical significance of viscosity is not a result of the work itself, but is a reflection derived from the results, together with the bibliography that supports the statements reflected here. Same with lines 124-128, 132-139, 154-156, 162-178, 202-205, 202-210, 221-223.

Lines 108-109. Seeing the low viscosity results of the compounded formulation, it is not expected to provide higher biopermanence than an aqueous simple solution of cysteamine, without viscosifying agents. In this work, it can be read that a higher viscosity is preferable in suspensions, although not in dissolutions, as is the case here. Cysteamine is a highly water-soluble molecule, from which it follows that its distribution is uniform throughout the formulation. As a basic concept of biopharmaceuticals, higher viscosities provide longer persistence time in topical application, a beneficial effect for ophthalmic administration since the bioavailability of the active ingredient is favored. Please, reformulate this paragraph. In addition, this paragraph has no bibliographic reference to support this theory. Please add them.

Lines 154-156. Please provide bibliographic support with which to justify that a higher concentration provides faster corneal absorption.

It would help the reader to integrate the results if Figures 7A, 7B and 7C are named in the body of the manuscript. Same with figures 5 and 6 when applicable.

Please provide a "Conclusions" section.

Author Response

Responses to Reviewer #2

This is a well written and easy to understand work comparing the main physical and chemical characteristics of two formulations of cysteamine, one commercial and the other prepared as a compounded formulation. Congratulations to the authors for their work in a field where the information available so far is limited, as is often the case with rare diseases.

Response: We are grateful for the reviewer's helpful comments, which are sure to improve the quality of the paper. Comments and questions are answered point-by-point. All changes in the manuscript had been marked with yellow.

  1. Most of the abstract is part of the "state of the art", but hardly any information is provided on the content of the paper. Please make reference to the tests that have been performed, as well as the main conclusions of the work.

Response: The abstract had been modified and extended according to the remark.

  1. In line 49, there are two commas instead of a quotation mark. Please correct.

Response: It had been corrected.

  1. Line 66 states that there are no approved presentations marketed in the European Union, although the Cystadrops presentation is approved in Europe (https://www.ema.europa.eu/en/medicines/human/EPAR/Cystadrops), although it is not marketed in all countries. However, it can be purchased as a foreign drug. 

Response: Thank you for the remark, we rephrased the sentence as follows: „Another disadvantage is the fact that Cystadrops® is only available as orphan drug since 2008 in the European Union and to overcome this problem as alternative ocular formula-tion magistral eye drops are prepared with various concentrations of CYA, using different excipients, and buffers”

  1. PAMPA meaning in figure 1 should be explained in the caption of line 97. 

Response: Caption of Fig.1 had been extended with definition of PAMPA according to the remark.

  1. Please, move the “Material and methods” section before “results”.With this, the material and methods would be item 2, the results would be item 3, and the discussion would be item 4.

Response: We did not take into consideration this change request, as editor indicated.

  1. One of the two formulations studied is a compounded formulation. Please describe in detail the steps followed for its preparation in the material and methods section. Based on what is the hydroxyethylcellulose percentage of 0.39% chosen? Are special sterile conditions required for its production? Does it start with sterile material? Does it work in laminar flow cabinets?

Response: The applied API and excipients were not sterile, but the preparation of the solution was conducted in laminar air flow cabinet under aseptic conditions consisting of four main steps:

  1. preparation of the buffer solution,
  2. mixing of hydroxyethylcellulose into a portion of the buffer solution to produce a viscous solution
  3. dissolution of the active substance and other excipients in the remaining buffer solution
  4. mixing of the two solutions followed by sterile filtration through a sterile polyethersulfone (PES) filter membrane of 0.22 µm pore diameter and 25 mm diameter (Stericup®, Merck-Millipore, Darmstadt, Germany).

The hydroxyethylcellulose concentration used in other magistral ophthalmic stock solutions, which is analogous to the viscosity of the tear film, was employed. The sterility of the solution is tested as per Ph. Eur.  Since the drug substance is sensitive to oxidation, plastic vials are filled with nitrogen bubbling in a laminar flow cabinet. No unit showed microbiological growth in the media fill tests on three batches.

 The main steps of the compounded eye drop preparation had been added to the section 4.2 in Materials and Methods according to the remark.

  1. This compounded formulation, with such a large number of components, is not feasible for routine preparation in pharmacy services. Please discuss this in the discussion section.

Response: The Section Discussion had been extended with the following:

Compounded formulations with several components can be prepared in the pharmacy following the official pharmacopoeial regulations in accordance with the principles set out in the resolution issued by the Council of Europe on the quality and safety assurance requirement for medicinal products prepared in pharmacies for the special needs of patients. If the active ingredients are available in the official pharmacopeia or the Authority provided permission for its particular use, compounding can be a promising alternative to substitute the medicine shortage. Enhancing the expanded production of compounded formulations beyond the scope of Good Manufacturing Practice (GMP) regulations is possible by establishing a risk-based framework. This framework should involve the direct oversight of healthcare professionals from Health Authorities, specifically focused on the circumstances of the manufacturing steps, including analytical support and manufacturing validation, and the physicians who should also approve therapeutic protocols before approving these medicines. This risk-based framework should be well-defined and regulate potential patient population (individual therapy planning) and production circumstances. It must be admitted that producing medicinal products outside of GMP puts a higher risk on the final products; however, the absolute lack of a particular product can put a higher risk instead on patients.

  1. Given the high oxidizability of cysteamine, what is the shelf life of the formulation if stored in a refrigerator? In RED and In vitro corneal permeability experiments, temperatures of 35 degrees are used for four hours. If you do not measure cystamine, the main degradation product of cysteamine, how do you know which part of the cysteamine is degraded and, for that reason, its passage through the membrane is not detected?

 Response: The storing stability of the eye drops had not been investigated in this study. However, no decomposition of CYA was observed during the RED investigations in the 4-hour time frame at 35oC. According to paper of Pescina et al (Pescina, S. et al Eur.J. Pharm. Biopharm., 2016, 107, 171-179.) the expected stability of the CYA-containing eyedrops could be approximately 1 week in refrigerator. The further development of magistral eye drop involving pH optimization and the addition of stabilizing agents (such as EDTA or Na-phosphate) could improve the shelf-life.

  1. Lines 107-108 should be part of the discussion, and it should be in this section that we comment on which of the two is more physiological. The text between lines 108-118 clearly corresponds to the discussion section, not the results. The clinical significance of viscosity is not a result of the work itself, but is a reflection derived from the results, together with the bibliography that supports the statements reflected here. Same with lines 124-128, 132-139, 154-156, 162-178, 202-205, 202-210, 221-223.

Response: All of the text sections marked by the reviewer contain specific experimental results, so their exclusion (relocation) would make this section inaccessible, while if we copied them into the discussion section, they would appear as redundant information in the manuscript. Thus, if possible, we would not comply with this reviewer's request in order to maintain the traceability of the manuscript. 

  1. Lines 108-109. Seeing the low viscosity results of the compounded formulation, it is not expected to provide higher biopermanence than an aqueous simple solution of cysteamine, without viscosifying agents. In this work, it can be read that a higher viscosity is preferable in suspensions, although not in dissolutions, as is the case here. Cysteamine is a highly water-soluble molecule, from which it follows that its distribution is uniform throughout the formulation. As a basic concept of biopharmaceuticals, higher viscosities provide longer persistence time in topical application, a beneficial effect for ophthalmic administration since the bioavailability of the active ingredient is favored. Please, reformulate this paragraph. In addition, this paragraph has no bibliographic reference to support this theory. Please add them.

Response: We clarified the paragraph and added relevant reference. See lines 112-119.

  • Lines 154-156. Please provide bibliographic support with which to justify that a higher concentration provides faster corneal absorption.

Response: We clarified our statement and provided relevant reference, See lines 160-162.

  • It would help the reader to integrate the results if Figures 7A, 7B and 7C are named in the body of the manuscript. Same with figures 5 and 6 when applicable.

Response: According to the comment, we named the Figures in the manuscript.

  • Please provide a "Conclusions" section.

Response: Section Conclusion had been added to the manuscript.